# Modeling Health Data Using Machine Learning Techniques Applied to Financial Management Predictions

Rafael Leon Sanz [1,*] and Pilar Leon-Sanz [2]

1 Automatic Department, Technical Superior School of Computer Engineering, University of Alcalá, 28801 Alcalá de Henares, Spain
2 School of Medicine, University of Navarra, 31008 Pamplona, Spain
* Correspondence: rafael.leon@coit.es

**Abstract:** Health management has steadily improved in performance and accuracy using IT technology. Hospitals and health institutions hold an enormous number of data in their software applications, which can be used with Big Data methodologies to extract useful information. One of the most challenging aspects of health institutional management is financial management; billing prediction is a key aspect to maintain a predictable service level for patients, avoiding unpleasant surprises and anticipating treasury management. Using patient data from public patient databases and applying a machine learning approach, this article offers a model that helps to make more precise and detailed financial plans.

**Keywords:** Big Data; Artificial Intelligence; Artificial Neural Networks; machine learning; health management



## 1. Introduction

One of the main elements of current welfare systems are health services, which represent one of the largest items in state budgets, so monitoring and predicting the necessary investment for medical and pharmaceutical services is a challenging and difficult task, but of great interest to both public and private institutions [1]. In the article, we propose to study how to improve the adoption of sound decisions in the management of hospital institutions, which includes a set of decisions aimed at achieving defined objectives to improve health provision and establish better methods to support these decisions using Artificial Intelligence methods [2,3].

We need new instruments because health services are extremely complex and expensive, especially since professionals require a high level of education and experience [4] and due to the expense of the necessary technical means: real estate, medical and surgical examination equipment, and food and laundry services, among others. In addition, these services must be provided 24 hours a day, 7 days a week. Health care costs grow every year, and financial management can be seriously affected by many circumstances. Events such as the COVID-19 pandemic can devastate even the best financial planning. Predictive tools are, therefore, crucial for optimal management and decision making, and as Cubillas et al. point out, this is especially important in the health sector. These authors focus on the analysis of the management of health resources in primary care centers [5]. However, it is an example of the interest in studying the issue in hospitals, more complex structures where it is especially important to have information in advance in order to anticipate and be able to optimize resources, which are always scarce but essential in improving patient care.

The search for models to predict the management and prices of financial markets is a highly researched topic, despite its complexity and the fact it is difficult to predict over time [6]. So, we propose applying similar processes and techniques already investigated

(such as machine learning) in health institutions, even though there are still few proposals on the issue [7].

Health services generate a large number of data, which in many cases can be used to extract higher-quality information, which allows better informed decisions to be made. A new approach using Artificial Intelligence (AI) can be used in this changing environment [8]. Modeling data and making predictions based on available data can help medical administrators and improve the decision process. AI and Big Data will likely become powerful differentiators for health institutions and will likely enable reaching new levels of precision in management, service improvement and user satisfaction. Hospital costs and resource management are widely discussed topics [1]. There are two main approaches for their analysis: bottom-up or top-down. The bottom-up method is generally more widely used and is a model where all hospital elements are objects (patients, services, etc.) and, in general, are structured hierarchically.

IT technologies have been an essential tool for managing all aspects of health services. Artificial Intelligence has a long history [9]. It is a set of advanced mathematical technologies, which create data models and apply sophisticated algorithms on these data, building ingenious solutions to solve difficult problems [10]. They can identify data patterns and classify them, which allows large amounts of information to be processed. Recently, it has being used with increasing success, covering many aspects and improving the control and output of processes. At present, the line of research in which AI does not intervene is rare. In this paper, we use Artificial Neural Networks (ANNs), to implement the proposed solution, because there are studies that have shown interest in applying this method to health data. Drewe-Boss demonstrated that with a Deep Neural Network, they could take advantage of the full complexity of patient records and outperform standard approaches [7].

ANNs are a set of computational elements that aim to emulate the functioning of the Neural Networks of our nervous system. Neural Networks have become extremely advanced and sophisticated and obtain very good results. They are associated with deep learning as well as machine learning and, in many cases, provide a solution where other algorithms cannot be applied. There are numerous tools and libraries to work with them.

The objective of the article is to propose better financial and resource management of a hospital using an Artificial Neural Network (ANN) in order to instantly calculate the projection of resources and billing that the hospital is expected to have in the short term.

In the next section (Materials and Methods), we review the methodology followed by considering some aspects of hospital financial management, and cost and billing analyses in a healthcare setting. We also discuss the types and characteristics of data that can be used in the proposed evaluation. Section 3 describes the proposed solution and the results. In Section 4, we draw some conclusions and propose future lines of work.

## 2. Materials and Methods

### 2.1. Hospital Financial Management and Operation

Hospitals provide health services via a complex organization that is reflected in hospital budgets [1]. It is common for economic management to be divided into four large blocks: personnel expenses, departmental budgets, future growth and extraordinary expenses [11]. The first three groups are negotiated and approved in advance, when the budget is prepared the previous year. Extraordinary expenses cover investments and contingencies that arise during the fiscal year, which can be difficult to avoid and are linked to the very essence of health services, especially with emergencies and epidemics.

In general terms, health costs can be defined as the valuation of the consumption of goods and services to obtain the products that encompass the medical care process, which include several groups of activities: outpatient and inpatient activities, extractive and diagnostic tests, surgeries, education, research, etc. Various types of costs can be attributed to each of these activities: direct costs, indirect costs, or structural costs; fixed and variable costs; etc. In addition, various accounting systems can be used. Moreover, financing

strategies to help to cover some or all of the costs involved in operating such institutions and to estimate the unit cost of primary care and hospitals have been researched [12].

Therefore, hospitals can be considered multi-product, multi-service activities, but it is not easy to calculate the cost of each of the activities carried out. It is not easy to evaluate the unit price of a simple X-ray, because the initial investment of the equipment, maintenance, energy consumption, consumables, salaries, amortization, etc., have to be considered. In other cases, such as hospitalization services, the costs are much more difficult to quantify. In addition, the same medical service has different prices, depending on where it is located. Moreover, each patient requires differentiated resources, even if they suffer from the same disease, depending on age, sex, allergies, concomitant chronic diseases, etc. Many factors affect the rise in health expenditures, such as high-cost medical technologies, changes in disease patterns and increasing demand for health services. All countries allocate a significant portion of resources to the health sector [12]. Consequently, managing the costs of health services is critical to maintaining services. Costs and billing are linked values.

*2.2. Health Data*

In this paper, which proposes a data management model, we reviewed public data collections from many sources, such as Spanish National Statistics Institute (INE), Eurostat, Centers for Medicare and Medicaid Services (CMS), Payroll-Based Journal Public Use Files, and the Payroll-Based Journal system. Daily resident census information was based on Minimum Data Set (MDS), Healthcare Research and Quality (HRQ), Kaggle website and many more. They are freely accessible data, and their use does not entail major ethical or legal problems.

Here, it is proposed, as is being done in many institutions, to process combined clinical and economic data. Listings of patients, their illnesses, and discharge data, or Diagnosis-Related Groups (DRGs), personal health records (PHR), results of medical examinations [7,13,14], provide valuable information for making short- and long-term resource planning predictions, because they can allow individual patient billing to be assessed in a hospital.

However, personal health data have the highest degree of protection, so accessing or working with these data is not easy [15,16]. One might ask: can an institution use administrative and health data of patients in its economic management? What requirements should be considered to access databases with personal health data of their patients?

In Europe, the official data protection regulation (EU) *2016/679* (General Data Protection Regulation (EU GDPR)) establishes that "the principles of data protection should apply to any information concerning an identified or identifiable natural person" [17]. Having compared five different legislative jurisdictions, Flaumenhaft and Ben-Assuli concluded that among the legislation reviewed, EU GDPR stands out as seemingly providing the most comprehensive and stringent protection measures; yet, it appears to leave significant room for interpretation and a degree of ambiguity in key areas [15].

In this situation, it can be argued that those responsible for the financial management of a hospital need access to personal data to be able to properly manage the institutions, since only this information allows financial and resource expenses to be analyzed and understood. In this aspect, it should be distinguished whether access to personal data is included in a healthcare project that is developed based on medical assistance or in a research project that seeks to improve the management of clinical expenditure in the short or mid-term [16].

It is clear that the professionals who make up the financial management services of a hospital can access, ethically and legally, personal data of an administrative nature for the management of the institution. It is not as clear that they can directly access these data, without specific consent, for a research project that aims to develop, for example, an algorithm that can be applied later.

Another ethical and legal consideration is the access to personal health data. It is possible to access the identified health data, after signing the corresponding data process-

ing contract, for all procedures related to clinical assistance to a person. However, for other uses, it is recommended to use anonymized health data, and if this is not possible, pseudonymized health data can be considered. "Personal data which have undergone pseudonymization, which could be attributed to a natural person by the use of additional information should be considered to be information on an identifiable natural person" [6]. For AI research and financial management, the pseudonymization of health data, together with a commitment not to identify individual persons, would make it easier to access patient data [7].

### 2.3. Artificial Neural Networks

Artificial Neural Networks (ANNs) are built by computing nodes that have several inputs and outputs. Each input has a weight that adjusts its value, and a bias is added to calculate the output. The design of the ANN is adapted to the problem to which it is applied, creating a model of the data. The operation of the Neural Network has several steps: a training phase, a verifying phase and an operational application. A good introduction to ANNs can be found in Oludare et al. [18].

Firstly, the ANN is trained with already classified data as inputs and outputs. With a backpropagation process and using a lost function to evaluate the gains obtained, the weights and the bias of each neuron are adjusted. The training data are presented to the network in a random fashion in batches. The training process is similar to the bootstrap statistical technique [19]. Formula (1) shows how the output of an individual neuron is calculated.

$$y_k = f\left(\sum_{i=0}^{i=n}(w_i * x_i) + B_k\right) \tag{1}$$

ANNs can have several layers, allowing more complex and more accurate calculations to be performed. The Neural Networks described so far can only have a linear behavior. An activation function is used to implement nonlinear arithmetic. As an example, we may see, in Equation (2), how the sigmoid function, applied to the output of the neural cell, generates a nonlinear output.

$$f(x) = \frac{1}{(1 + e^{-x})} \tag{2}$$

In the training phase, a loss function is used to evaluate the accuracy of the predictions. We used the Mean Squared Error (MSE) function, which is a simple and commonly used technique [20]. The MSE formulation can be seen in Equation (3).

$$mse = \frac{1}{N}\sum_{i=1}^{N}(y_i - \hat{y}_i)^2 \tag{3}$$

To calculate the optimization of the network parameters, optimizer functions are used. Stochastic Gradient Descent (SGD) has been widely adopted in machine learning for many years. A good description can be found in Bottou [21]. We used the Adam optimizer function, introduced by Kingma and Ba [22], which has a good performance result in problems that involve a large number or parameters, such as Neural Networks.

As stated above, when the ANN results are inside the precision criteria previously established, a verification phase is accomplished, where a portion of new data already classified are presented as input and the output is verified. If the tests satisfy the precision criteria, the ANN can start its production life and can be used with real data.

We chose as a first approach the use of a Neural Network very similar to the original Perceptron. A Perceptron uses the Sequential Model, which is a plain stack of neural layers, where each layer has a tensor as input and as output. Subsequently, a Deep Neural Network (DNN) was used. DNNs have hidden layers, and they have more computing power; a good introduction to and the evolution of DNNs can be found in Samek and Montavon [23]. In further experiments, new Neural Network architectures can be tested.

*2.4. Testing Environment*

The three main elements that build the solution are data, software and the computing device, and a cloud solution could be recommended. We used one data collection with individual DRG patient information from the Kaggle website [24], the Inpatients Charges Data collection, with 163,066 rows and 12 columns. Most of these data are associated with the corresponding hospital and its location. We focused on Diagnosis-Related Group, Average Covered Charges (ACC), Average Total Payments (ATP) and Average Medicare Payments (AMP). We chose to predict Average Total Payments from the other columns. For this, we trained the neural networks with part of the data collection, selecting 60% of each set of DRG cases. We used another 20% of the DRG sets to test the results. Finally, we tested the predictions of the trained Neural Network with the rest of the 20% of various DRG data sets, and we present these results in the next section.

This data collection is limited for an exhaustive demonstration, but it is sufficient to show that our proposal is feasible. In addition, any hospital has access to this kind of information in their IT systems.

With respect to the software environment, we used Tensorflow, a Google Artificial Intelligence Product, which is an open-source, end-to-end machine learning platform [25]. Tensorflow incorporates several tools, such as Keras [26], as a facilitator layer, and Pandas [27], as a data management tool. We used Python [28] as the programming language, for its versatility and efficiency. As a computing device, we used an Intel computer with satisfactory results. In future work, cloud computing should be tested.

To complete the verification of the methodology, it seemed necessary to establish some type of comparison with some other technique. Choosing the appropriate technique for a comparison is not trivial, as each one has its peculiarities, and often, a data adaptation process is necessary. Therefore, to complement the analysis of the test environment, it is convenient to apply an additional technique to make sure which is the best method to use. Consulting the literature, we compared the Neural Network with the Decision Tree [29].

## 3. Results

In this section, we assess the computational results of the proposed solution. The purposes of these tests were (i) to demonstrate that the solution exists and that the predictions are possible to be performed and (ii) to show that the computation time allows one to have updated data results every day. The longest time is spent training the network, a process that is not needed daily.

All the experiments were performed with a computer with an Intel I7-9750H CPU processor with a clock signal range of 2.6–4.5 GHz, with 16 GB of RAM, with one Intel UHD GPU and an NVIDIA GeForce GTX GPU CUDA enabled, running Windows 10. The ANN code ran on Tensorflow and Keras version 2.10.0 and Pandas version 1.4.3. We tested several configurations, essaying with different numbers of layers, activation functions, sizes of batches and numbers of epochs.

Because the data were not numerous, training and execution time were under 600 s in all cases. Improving performance can be considered as a future area of research. Real data collections are much bigger, and it is an effort that cannot be underestimated.

The Neural Network results using the Inpatients Charges Data collection are presented in the next section.

*3.1. Training, Test and Results of Simple Perceptron Neural Network*

As mentioned, we used two different models to process the data: a simple Perceptron implemented with the Sequential Neural Network Model, which is the most basic approach, and a more sophisticated and advanced design that uses several layers, known as Deep Neural Network (DNN). In this section, we present the results of the Perceptron. In Figure 1, the Perceptron architecture can be seen.

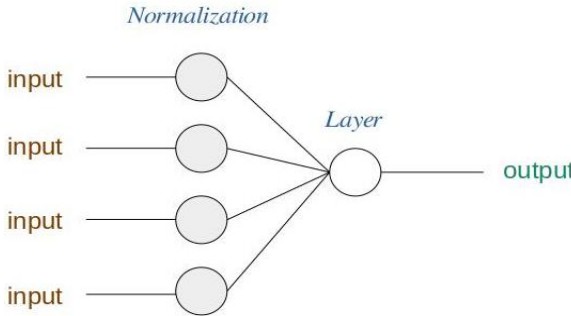

**Figure 1.** First Neural Network design. Prepared by the authors.

This ANN has a simple structure: one normalization layer and one processing layer, with a total of five parameters. It uses Mean Absolute Error as the loss function and Adam as the optimizer function. The training process used 150 epochs with batch sizes of 50 cases. It took 5 min, 55 s and 930 milliseconds. In Figure 2, it is possible to see the evolutions of the values of the loss function during the training process. In the first 60 epochs, the process improved the adaptation of the network results. From that moment, the improvement was negligible.

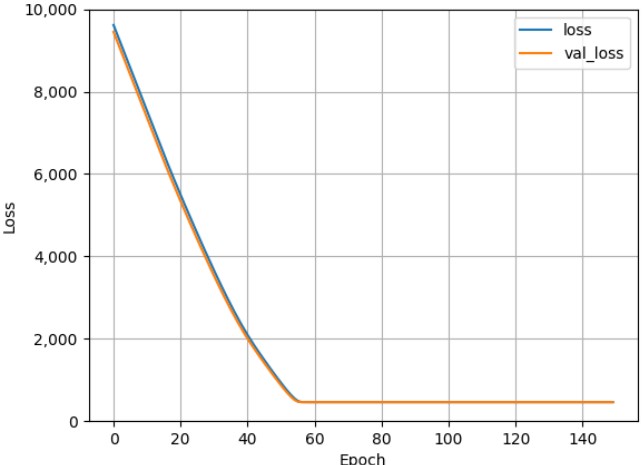

**Figure 2.** Sequential Neural Network training evolution. Prepared by the authors.

The dataset has data of patients that belong to 100 different DRGs. We tested the Neural Network on a set of DRGs of patients chosen randomly, predicting their Average Total Payments. We selected seven of them, and we present the results obtained in Table 1. The columns of this table are the DRG code, the number of patients that belonged to the diagnostic group, the time spent by the network to make the predictions and the accuracy achieved by the model for each group.

**Table 1.** Sequential Neural Network results on DRG billing prediction.

| DRG | Number of Patients | Time | Test Accuracy |
|---|---|---|---|
| 39 | 1079 | 3.957 s | 85.04% |
| 64 | 1659 | 4.344 s | 79.48% |
| 178 | 1953 | 4.250 s | 83.68% |
| 473 | 846 | 4.188 s | 83.55% |
| 536 | 1142 | 4.480 s | 83.75% |
| 690 | 2989 | 4.105 s | 83.04% |
| 811 | 1135 | 4.169 s | 80.14% |

The accuracy achieved was around 80%, and the time spent running the Neural Network was around 4 s. It can be concluded that the Neural Network has a good prediction capacity and that the time for running is acceptable. We were not able to find a relation between the number of cases in each group and the accuracy value or the execution time.

In Figure 3a,b, the prediction results and error count for a particular DRG, group 178, are plotted.

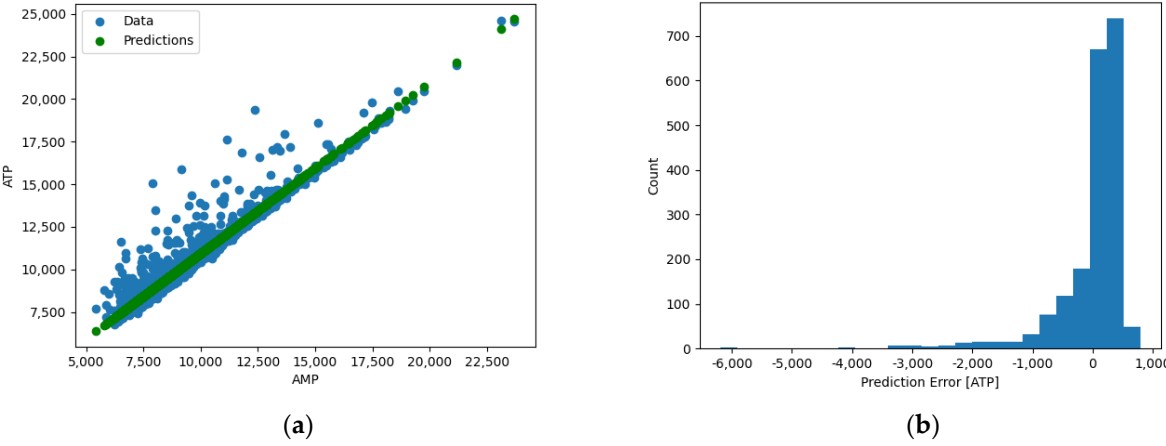

**Figure 3.** Perceptron results for DRG 178: (**a**) results using AMP to predict ATP and (**b**) error count. Prepared by the authors.

As a final comment, the simple Sequential Model had good performance, and it was mainly because the relationship among the variables was linear. This model is a simple and effective approach to solve the problem under the conditions defined, but we do not believe that it could be used in real applications.

### 3.2. Training, Test and Results of DNN

In this section, the Deep Neural Network results are presented. The structure of this Neural Network has one normalization layer, two hidden layers and one output layer, with a total of 4,488 parameters. It uses Mean Absolute Error as the loss function, Adam as the optimizer function and Rectified Linear Unit (Relu), as the activation function. In Figure 4, it is possible to see its architecture.

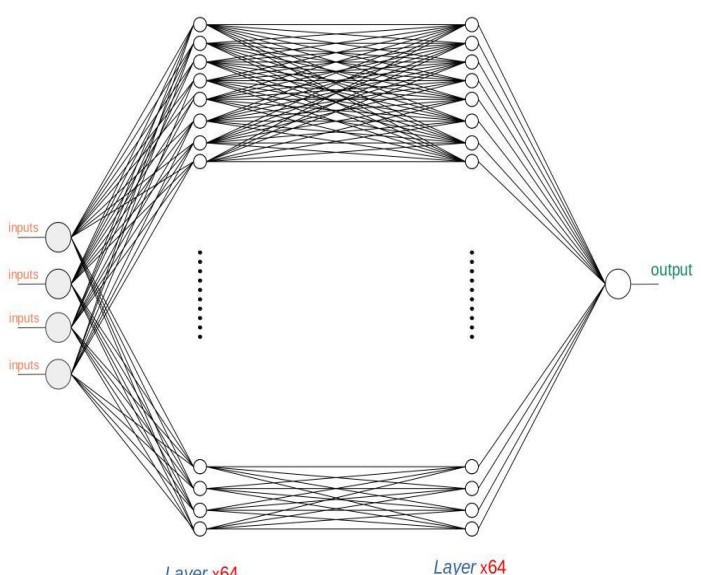

**Figure 4.** Deep Neural Network architecture. Prepared by the authors.

The network training process was configured with 150 epochs with batch sizes of 50 cases. It took 6 min, 16 s and 968 milliseconds to train the DNN. In Figure 5, it is possible to see the evolution of the values of the loss function during several training processes.

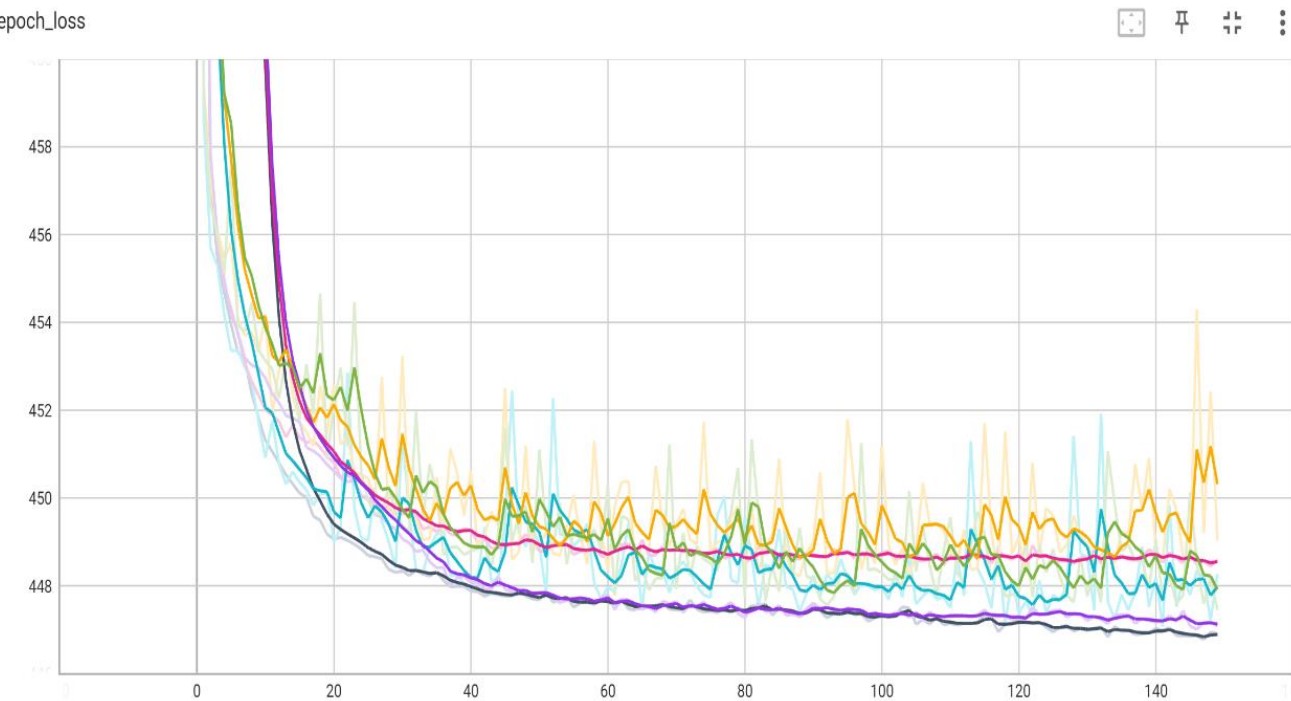

**Figure 5.** DNN Neural Network training evolution (from Tensorboard).

It can be seen that the training process evolved quickly for the first 50 epochs and that, after that, the improvement was slight. In order to compare the results, we tested the same DRG sets as in the previous case. The table of executions for these DRGs can be seen in Table 2, which has the same columns as that in the previous case, i.e., the DRG code, the number of patients that belong to the diagnostic group, the time spent by the network to make the predictions and the accuracy achieved by the model for each group.

**Table 2.** DNN Sequential Neural Network results on DRG billing prediction.

| DRG | Number of Patients | Time | Test Accuracy |
|-----|-----|-----|-----|
| 39 | 1079 | 4.490 s | 85.08% |
| 64 | 1659 | 4.439 s | 79.48% |
| 178 | 1953 | 4.415 s | 83.68% |
| 473 | 846 | 4.405 s | 83.55% |
| 536 | 1142 | 4.415 s | 83.75% |
| 690 | 2989 | 4.467 s | 83.04% |
| 811 | 1135 | 4.379 s | 80.14% |

This Neural Network also had a good prediction capacity, very similar to that of the Perceptron. The accuracy achieved was around 80%, slightly higher than before; the time spent running the Neural Network was slightly longer, between 4 and 4.5 s. We also plotted the results for DRG 178 and the count of the value of the errors. They can be seen in Figure 6a,b.

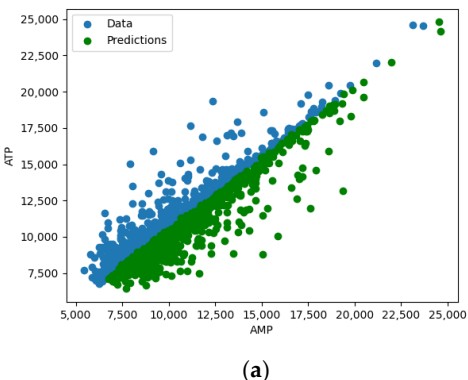

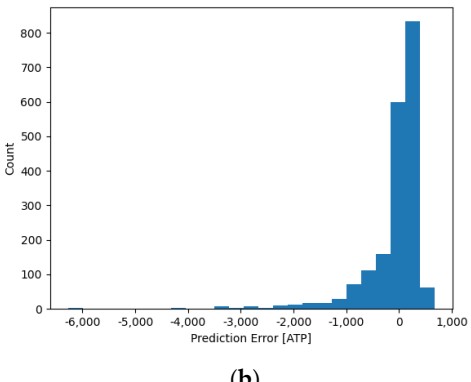

(**a**)

(**b**)

**Figure 6.** DNN predictions and errors for DRG 178: (**a**) results of the predictions (**b**) errors count.

To have a better understanding and compare the performance of both Neural Networks, in Table 3, a comparison between both models in terms of their accuracy and time spent for every DRG can be seen.

**Table 3.** Perceptron Neural Network results vs. DNN results.

| DRG | Perceptron Time | DNN Time | Perceptron Accuracy | DNN Accuracy |
|---|---|---|---|---|
| 39 | 3.957 s | 4.490 s | 85.04% | 85.08% |
| 64 | 4.344 s | 4.439 s | 79.48% | 79.48% |
| 178 | 4.250 s | 4.415 s | 83.68% | 83.68% |
| 473 | 4.480 s | 4.405 s | 83.55% | 83.60% |
| 536 | 4.105 s | 4.415 s | 83.75% | 83.75% |
| 690 | 4.105 s | 4.467 s | 83.04% | 83.04% |
| 811 | 4.169 s | 4.379 s | 80.14% | 80.14% |

A slightly better performance of the DNN was observed, at the cost of some processing time.

### 3.3. Comparison with Decision Tree Technique

In order to further evaluate the results obtained, as we explain in the methodology section, we used a Decision Tree as another technique using the same data.

To implement the Decision Trees, we used the Knime application [30], an open-source data analytics tool. This application implements a wide set of techniques with good results, including Decision Trees. The Knime Decision Tree implementation output is categorical, so we adapted patients' data to this circumstance. We grouped the Average Total Charge column in eight categories.

In Table 4, the list of categories can be seen.

**Table 4.** Range of predicted values of the ATC variable.

| Category | Billing Range Category |
|---|---|
| 1 | value < 3500 |
| 2 | 3500 < value < 5000 |
| 3 | 5000 < value < 7000 |
| 4 | 7000 < value < 9000 |
| 5 | 9000 < value < 10,500 |
| 6 | 10,500 < value < 15,000 |
| 7 | 15,000 < value < 30,000 |
| 8 | value > 30,000 |

The tests were implemented by building a Decision Tree with the same training data used to train the Neural Networks and later by applying this Decision Tree to the same patient DRGs. In Table 5, the accuracy obtained is shown compared with the Neural Networks.

**Table 5.** Decision Tree results and comparison with Sequential Model and Deep Neural Network.

| DRG | Linear ANN Accuracy | DNN Accuracy | Decision Tree Accuracy |
|---|---|---|---|
| 39 | 85.04% | 85.08% | 80.35% |
| 64 | 79.48% | 79.48% | 66.67% |
| 178 | 83.68% | 83.68% | 68.40% |
| 473 | 83.55% | 83.55% | 59.57% |
| 536 | 83.75% | 83.75% | 72.68% |
| 690 | 80.04% | 80.04% | 67.27% |
| 811 | 80.14% | 80.14% | 68.10% |

In a first approach, both Neural Networks and the Decision Tree achieved admissible results. For these data, the Neural Network accuracy was around 80%, and that of the Decision Tree was around 70%. The main goal of this comparison was not to make a thorough performance evaluation or create a competition between different AI methods; the goal was to make an assessment of whether applying machine learning to hospital financial management data is possible, whether more than one technique can be used and whether their results provide more confidence in these techniques.

## 4. Discussion

As Dash et al. explain, an efficient management, analysis and interpretation of health Big Data can open new avenues for modern healthcare [14]. Considering our results, we think it is time to take vigorous steps to integrate the analysis of biomedical, healthcare and economic data for better financial management of healthcare organizations.

Otherwise, although personal health data in general have been considered a promising tool for the management of the healthcare system as a whole, widespread acceptance has yet to be achieved. A significant obstacle often cited as delaying the implementation of AI systems is the concern regarding the ability to properly ensure the security and privacy of this sensitive information [15].

There are already several well-known commercial solutions in hospitals, such as Enterprise Resource Planning (ERP), with powerful features to connect and manage all the processes that occur in a hospital, from the providers to the end users, including financial management. Those tools allow every aspect of a health service, past or present, to be analyzed in detail. However, they mainly rely on managers' experience to make future projections.

It is also possible to build tailor-made software, using algorithmic solutions to process data to help managers to make more informed decisions. The inconvenience of this kind of solutions is that they need to figure out hundreds of rules that link patients' illnesses with the resources needed for their treatments. Because all this information changes and new treatments appear every year, evolving this set of rules could be a cumbersome job. On the other hand, working with Neural Networks, in most cases, implies just new training with the new data and testing their results.

There is a growing interest in applying machine learning techniques to financial forecasting in the health services environment. A good example is the research study published in the present year (2022) by Drewe-Boss et al. [7]. In this study, machine learning was applied to predict the costs that a patient would incur based on their diagnosis. They used Deep Neural Networks using assurance companies' data. The main difference with

respect to our article is that their target was costs prediction and our work is focused on billing forecasting. Moreover, their scope of work was insurance companies, while we are focused on the hospital level. Both research efforts reflect the strength of Artificial Intelligence in this area.

Gomez-Cabrera and Escamilla-Ambrosio also recognized that current machine-learning algorithms implemented in building structural health monitoring systems are successful tools for health care systems and the care of patients and that the specific application of ANNs improves the reliability and performance of the systems [31].

Neural Network software has evolved and matured; it has reliable outputs with consistent environment and tools. We tested two designs: a basic Sequential Model and a more advanced Deep Neural Network with similarly good results in both cases, but with a slight advantage of the DNN, at a price of more complexity and longer execution times. At a first glance, it could be thought that it is not worth using it, but a deeper analysis showed that the DNN predictions better fitted the shape of the data. The Sequential Model is of little use if the relation between the variables is not linear, which can be verified by comparing Figures 3a and 6a. Therefore, the use of DNNs should be recommended.

We also verified our hypothesis by repeating the predictions using a Decision Tree. The results were comparable, with Neural Networks being slightly better. This underpins the use of machine learning techniques in the financial environment of hospital management.

One limitation of this study comes from the lack of standards in how each hospital organizes their data sets. Therefore, the proposed model requires an adaptation to each of the hospitals where it is applied. This also has repercussions on the design of the Neural Network, that would need reshaping and retuning. These additional efforts may represent a small barrier to the deployment of the project in new hospitals, although, on the other hand, they may also entail a customization that improves the performance of the solution in each particular case.

As future research developments, new designs of Neural Networks can be proposed. Pharmaceutical planning and forecast could be another interesting field of application of this methodology.

Another field of research could be the historical hospital budget chapters using Recurrent Neural Networks that are very good in serial number predictions. With hospital historical data on economic budget and summary of budget execution, it could be another way to make accurate predictions from monthly and quarterly perspectives.

## 5. Conclusions

The article supports the use of machine learning techniques in the financial environment of hospital management. The research study presents a new efficient way to predict short-term financial costs in hospitals using an Artificial Neural Network. Hospital Big Data and AI are an exceptional field of research to help hospital administrators in their resource planning decisions. We show that the large collections of data stored in hospitals can be transformed into knowledge, which could result in the improvement of the quality of health services and medical care of patients. It is necessary to design instruments so that administrative services can use health data in real time, in an ethical and legal way, for administrative and financial purposes.

From the compared proposals, we could conclude that the DNN predictions better fitted the shape of the data, because the simple Sequential Model is useless if the relationships between the variables are not linear. We also verified our hypothesis using a Decision Tree. Moreover, although the results were similar, we recommend using DNNs, because Neural Networks obtained better and more accurate results.

We were able to verify that there are recent initiatives in this field. The existence of research groups presents a great opportunity for multidisciplinary teams to collaborate in the academic arena in order to move forward by considering all aspects involved.

**Author Contributions:** Conceptualization, R.L.S. and P.L.-S.; Methodology, R.L.S.; Software, R.L.S.; Validation, R.L.S.; Formal analysis, R.L.S.; Investigation, R.L.S.; Resources, R.L.S. and P.L.-S.; Data

curation, R.L.S.; Writing—original draft, R.L.S. and P.L.-S.; Writing—review & editing, R.L.S. and P.L.-S.; Visualization, R.L.S. and P.L.-S.; Supervision, P.L.-S.; Project administration, P.L.-S. All authors have read and agreed to the published version of the manuscript.

**Funding:** This research received no external funding.

**Institutional Review Board Statement:** Not applicable.

**Informed Consent Statement:** Not applicable.

**Data Availability Statement:** Original Data could be found at: https://www.kaggle.com/datasets/speedoheck/inpatient-hospital-charges accessed on 20 October 2022.

**Conflicts of Interest:** The authors declare no conflict of interest.

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
