# Peer review of "Modeling Health Data Using Machine Learning Techniques Applied to Financial Management Predictions"

_applsci, doi:10.3390/app122312148_

Round 1

Reviewer 1 Report

Thank you for the opportunity to read and review your paper entitled “Modeling Health data using Machine Learning technics applied to financial management predictions”.

In my opinion, the topic is on time and the paper can contribute to the current academic debate. I hope you will find the comments useful to further improve the paper. 

I look forward to receiving your revised manuscript soon.

Title

The title, “Modeling Health data using Machine Learning technics applied to financial management predictions” expresses at its best the content of the paper. 

Abstract

The abstract full expresses the aim, the theory and results which increase reader’s interest in keep reading the contribution. 

Originality: Does the paper contain new and significant information adequate to justify publication?

The paper addresses an important research question which is under the current academic debate. Hence, I believe that the paper can significantly contribute to the extant literature.  

Introduction

In my opinion, some parts of the “Introduction” section are too short and not well connected with other parts. The author(s) should link them better. In addition, the author(s) should better describe how the current paper contributes to the extant literature. 

Material and Methods

The “Materials and Methods” section is generally well organised. 

However, I ask to the author(s) to improve this section by providing more insights on recent articles on the same topic published in this journal, if available.

Practicality and/or Research implications

The author(s) should evaluate the opportunity to stress further the implication for practice and academic world, since they appear to be too weak.

Conclusion

The author(s) should enlarge the "Conclusion" section, since it appears too short in the current state.

Quality of Communication

The overall quality of the paper is adequate, since the author(s) used the proper technical language of the field. However, it is possible to notice a strict separation among the various section of the paper. Although the structure of the paper is clear, the logical steps that lead from a section to another section are not clear too. The author(s) should try to better link the various sections of the manuscript. 

In addition, I recommend to author(s) to check carefully potential grammar errors before to submit the revised manuscript, as well as to make the writing more fluid, careful and clean. The use of a proofreading service is strongly recommended.

Author Response

First of all, the authors want to thank the reviewers’ comments and suggestions on the article. We have made all the changes suggested.

We have reviewed with an expert in proofreading English, language, style and spelling.

Title: We would like to point out that we have corrected a misspelling of a word in the title.

Introduction

We have enlarged the “Introduction” section and connected it better with other sections of the article.

We have also included some references as well as explained how the current paper contributes to the extant literature.

Material and Methods

The authors have improved this section providing more insights on recent articles on the same topic published in this and other journals.

Practicality and/or Research implications

We have shown in the Discussion some examples of the implications of the research for practice and the academic world.

Conclusion

We have enlarged the "Conclusion" section and integrated the results obtained and explained in the Results section.

Reviewer 2 Report

The manuscript presents research related to improving financial management in health care based on the application of artificial intelligence techniques. The set goal was realized using machine learning. The advantages of the proposed financial flow management method are proven through modeling results and numerical experiments.

In general, the work is on a current topic, has good potential for development, and my overall assessment is positive. My main observations and comments are as follows:

- as notations and abbreviations are used in the text, I recommend authors to add a list of used notations at the beginning or end of the manuscript.

- I would recommend the authors review the quality of all figures (figure 1 is of very poor quality), and also the formatting and editing of the text;

- a more detailed report on the state of the research problem should be made, containing more relevant literature sources, and also a comparison of the obtained results with those obtained by using other techniques;

- it would be useful, in addition to the advantages in the conclusion section, for the authors to comment on the disadvantages of the proposed method for managing financial flows. In this way, a complete picture of its qualities, capabilities and application limitations will be obtained. Moreover, there is a lack of in-depth analysis of the obtained results and it is not clear what conclusions and conclusions should be drawn on this basis;

- I recommend describing in detail the methodology and the related algorithm for implementing machine learning, so that the presented approach can be more useful and accessible to readers.

Author Response

Authors’ Notes to Reviewer

First of all, the authors want to thank the reviewers’ comments and suggestions on the article. We have made all the changes suggested.

We have reviewed with an expert in proofreading English, language, style and spelling.

Title: We would like to point out that we have corrected a misspelling of a word in the title.

Introduction

We have enlarged the “Introduction” section and connected it better with other sections of the article.

We have also included some references as well as explained how the current paper contributes to the extant literature.

Material and Methods

The authors have improved this section providing more insights on recent articles on the same topic published in this and other journals.

Practicality and/or Research implications

We have shown in the Discussion some examples of the implications of the research for practice and the academic world.

Conclusion

We have enlarged the "Conclusion" section and integrated the results obtained and explained in the Results section.

Other considerations

- We have limited and clarified the abbreviations in the text

- We have revised and improved the quality of all figures, especially figure 1

- We have expanded in more detail the status of the research problem, with more relevant bibliographical references.

- We have described the methodology in more detail. We hope that it can thus be more useful and accessible to readers.

- We have also discussed the results obtained with those obtained by other techniques.

- We have added the limitations of the study and the method proposed in the Discussion

- We have completed the analysis of the results obtained.

- We have expanded the conclusions.

Round 2

Reviewer 2 Report

The authors have revised the manuscript, largely following my comments and recommendations. I believe that, in order to make the manuscript easier to work with, it is necessary to explain the symbols used in the text or to distinguish them at the beginning or end.

Author Response

Thank you very much for the very favorable comment about the changes we introduced in the manuscript after your first report.

As authors, we really appreciate your comments and suggestions. We also think that the article has improved significantly.

Regarding the only question pending: “in order to make the manuscript easier to work with, it is necessary to explain the symbols used in the text or to distinguish them at the beginning or end”. We don't quite understand the question, but in any case, we have verified that we have explained the meaning of each acronym that appears in the text.

And regarding the three formulas included (the only symbols we have used), although they are basic, we have also verified that they are explained in the text.

We have also checked other recent articles published by Applied Sciences and the authors follow the same system. For example:

- Mateo-March, M.; Lillo-Beviá, J.R.; della Mattia, G.; Muriel, X.; Barranco-Gil, D.; Zabala, M.; Courel-Ibáñez, J.; Salas-Montoro, J.A. Power Profile Index: An Adjustable Metric for Load Monitoring in Road Cycling. Appl. Sci. 2022, 12, 11020. https://doi.org/10.3390/app122111020 (“Highly Accessed Articles”)

- Dragusanu, V.; Lunguleasa, A.; Spirchez, C. Evaluation of the Physical, Mechanical, and Calorific Properties of Briquettes with or without a Hollow Made of Wheat (Triticum aestivum L.) Straw Waste. Appl. Sci. 2022, 12, 11936. https://doi.org/10.3390/app122311936

- Etc.

We think we have finished the review, please feel free to let us know if we need to add anything else.

Acronym used in the article:

Artificial Intelligence (AI)

Artificial Neural Networks (ANN)

Spanish National Statistics Institute (INE)

Centers for Medicare and Medicaid Services (CMS)

Minimum Data Set (MDS)

Healthcare Research and Quality (HRQ)

Diagnosis-related groups (DRGs)

Personal health records (PHR)

General Data Protection Regulation (EU GDPR)

Mean Squared Error function (MSE)

Stochastic Gradient Descent (SGD)

Deep Neural Network (DNN)

Average Covered Charges (ACC)

Average Total Payments (ATP)

Average Medicare Payments (AMP)

Rectified Linear Unit (Relu)

Enterprise Resource Planning (ERP)
